Optimal extraction, purification and antioxidant activity of total flavonoids from endophytic fungi of Conyza blinii H. Lév

Zhao Shuheng 1
Wu Xulong 2
Duan Xiaoyu 1
Zhou Caixia 1
Zhao Zhiqiao 1
Chen Hui 1
Tang Zizhong 14126@sicau.edu.cn 1
Wan Yujun 3
Xiao Yirong 4
Chen Hong 5
1 College of Life Sciences, Sichuan Agricultural University , Ya’an , China
2 Chengdu Agricultural College , Chengdu , China
3 Sichuan Food Fermentation Industry Research and Design Institute , Chengdu , China
4 Sichuan Agricultural University Hospital , Ya’an , China
5 College of Food Sciences, Sichuan Agricultural University , Ya’an , China
Mora-Montes Hector
Electronic publication date: 2021 Apr 9
Publication date: 2021
Volume: 9
Electronic Location ID: e11223
Received 2020 Dec 10; Accepted 2021 Mar 15
Copyright: ©2021 Zhao et al.
Copyright year: 2021
Copyright holder: Zhao et al.
License: This is an open access article distributed under the terms of the Creative Commons Attribution License, which permits unrestricted use, distribution, reproduction and adaptation in any medium and for any purpose provided that it is properly attributed. For attribution, the original author(s), title, publication source (PeerJ) and either DOI or URL of the article must be cited.
License URL: https://creativecommons.org/licenses/by/4.0/

Keywords: Chaetomium cruentum, Response surface, Macroporous resin, Flavonoid

Funding: Application Fundamentals (Key) R&D Project of Sichuan Provincial Science and Technology Department 2019YJ0549 Seedling Project Cultivation Project 2019058 Key R&D Project of Sichuan Provincial Department of Science and Technology 2019YFG0154 This work was supported by the Application Fundamentals (Key) R&D Project of Sichuan Provincial Science and Technology Department (Grant No. 2019YJ0549), Seedling Project Cultivation Project (Grant No. 2019058) and the Key R&D Project of Sichuan Provincial Department of Science and Technology (Grant No. 2019YFG0154). The funders had no role in study design, data collection and analysis, decision to publish, or preparation of the manuscript.

==============================
Background

Flavonoids are widely used in the market because of their antibacterial, antiviral, and antioxidant activities. But the production speed of flavonoids is limited by the growth of plants. CBL9 (Chaetomium cruentum) is a flavonoid-producing endophytic fungi from Conyza blinii H. Lév, which has potential to produce flavonoids.

Methods

In this study, we isolated total flavonoids from endophytic fungus CBL9 of Conyza blinii H. Lév using macroporous resin D101. The process was optimized by response surface and the best extraction process was obtained. The antioxidant activities of total flavonoids were analyzed in vitro.

Results

It was found that the best parameters were 25 °C pH 2.80, 1.85 h, and the adsorption ratio reached (64.14 ± 0.04)%. A total of 60% ethanol was the best elution solvent. The elution ratio of total flavonoid reached to (81.54 ± 0.03)%, and the purity was 7.13%, which was increased by 14.55 times compared with the original fermentation broth. Moreover its purity could rise to 13.69% after precipitated by ethanol, which is very close to 14.10% prepared by ethyl acetate extraction. In the antioxidant research, the clearance ratio of L9F-M on DPPH, ABTS, •OH, •O2−, (96.44 ± 0.04)% and (75.33 ± 0.03)%, (73.79 ± 0.02)%, (31.14 ± 0.01)% at maximum mass concentration, was higher than L9F.

Conclusion

The result indicated using macroporous resin in the extraction of total flavonoid from endophytic fungus is better than organic solvents with higher extraction ratio, safety and lower cost. In vitro testing indicated that the flavonoid extracted by macroporous resin have good antioxidant activity, providing more evidence for the production of flavonoid by biological fermentation method.

Introduction

Flavonoids are important secondary metabolites of plants, which contain diverse pharmacological activities owning to complex structure types (Yonekura-Sakakibara, Higashi & Nakabayashi, 2019). For example, flavonoids have a strong antioxidant effect on blood circulation and cardiovascular system (Echeverría et al., 2017). Calycosin have significant antiviral activity both in vivo and vitro (Zhu et al., 2009). Most flavonoids have a significant inhibitory effect on the growth of bacteria including Bacillus subtilis, Staphylococcus aureus and Escherichia coli (Kamrani et al., 2007; Gadkowski et al., 2019).

Endophytic fungi widely exist in advanced plant. It has obvious host specificity and tissue specificity (Martin, Romina & Priscila, 2013; Carroll & Carroll, 1978). Endophytic fungi is able to produce the same or similar secondary metabolites of the host, including flavonoid with excellent activity (Qiu et al., 2010; Shih et al., 2017; Shou-Jie et al., 2018; Chi et al., 2019). CBL9 is a flavonoid-producing endophytic fungus from Conyza blinii H. Lév, which belongs to Chaetomium and is used to produce flavonoid with excellent antioxidant effects in vitro (Tang et al., 2020).

Macroporous resins have been used in the extraction of flavonoids widely as a result of its advantages (Du et al., 2012; Li, Chen & Di, 2012; Chen et al., 2013; Zhang et al., 2018). But macroporous resin has not been used in the endophytic fungi of Conyza blinii H. Lév. currently. The macroporous resin D101 was used to extract the total flavonoids of CBL9 for further uses, and the response surface method was optimized to obtain the best process to extract the total flavonoid of the endophytic fungi of Conyza blinii H. Lév.

Materials and Methods

Plant material

The endophytic fungus CBL9 separated from Conyza blinii H. Lév: Biochemistry and Molecular Biology Laboratory of Sichuan Agricultural University preserved

Chemical reagents

2, 2-Diphenyl-1-picrylhydrazyl (DPPH, HPLC) was purchased from Yuanye Biotechnology Co. (Shanghai, P. R. China). Ferrous Sulfate Heptahydrate (FeSO4•7H2O), hydrogen peroxide (30%H2O2), Pyrogallol and Concentrated hydrochloric acid were purchased from Xilong Chemical Co. (Sichuan, P. R. China). Ascorbic acid and Vitamin C (Vc, AR) were purchased from Sinopharm Chemical Reagent Co. (Shanghai, P. R. China). Other chemicals and solvents used in this study were analytical grade.

Study on optimization of extraction process of crude flavonoid

Preparation of raw mater

The flavonoid-producing endophytic fungus CBL9 was inoculated into fresh PDA medium and cultured with shaking at 28 °C until the mycelial pellets grew to a certain condition (Bacteria is not growing and the concentration of flavonoids in the fermentation broth reaches 10 mg/L), then fermentation broth was rotary evaporation at 50 °C and raw material (L9F) was obtained after freeze-drying.

Ethyl acetate extraction

The extraction was conducted by following the method of Saraswaty with slight modifications (Saraswaty et al., 2013). A total of 200 mL concentrated fermentation broth was mixed with two times the volume of ethyl acetate to extract the fermentation broth, and repeat extraction three times. Crude flavonoid (L9F-E) was obtained by concentrating by evaporation under freeze-drying.

Preparation of standard curve

Sodium nitrite-aluminum nitrate colorimetric method was used to draw rutin standard song by following the method of Tang et al. (2020). The standard curve equation: A = 0.4164C-0.0003 (R2 = 0.999 4).

Single factor test

The single factor test method was conducted with slight modifications (Bi & Tan, 2012) and the Macroporous resin D101 was used (Zhang et al., 2018). First, 1.0 g wet resin pretreated was put into 15 conical flasks and mixed with certain L9F. Then the mixture was incubated at different temperature, pH for different time. The adsorption ratio of flavonoid was measured.

Response surface analysis

According to the single-factor study, the Box-Benhnken optimization study was designed (Yu et al., 2019) with three levels (adsorption time, temperature, and pH). The adsorption ratio was chosen as inspection index. Design-Expert was used to analyze based on Box-Benhnken data. The possible mathematical model is: Y=β0+Σβixi+Σβijxixj+Σβijxi2

Y is the predicted response value; β⋅0, βi, βii and βij is the regression coefficient representing the interaction of intercept, linear, squared and two factors; xi and xj is the independent factor of encoding (i ≠ j).

Influence of eluent concentration

Gradient volume fraction of ethanol-water solution was used to elute saturated adsorbent at a certain flow rate (1 mL/min). its purity and elution ratio was measured. The total flavonoid (L9F-M) was extracted under optimal condition we got in 2.3.5 and 2.4.

In vitro antioxidant activity assay

DPPH radical scavenging assay

The DPPH radical scavenging activities was collected as previously described in Kaur et al. (2020) and Tsimogiannis & Oreopoulou (2006). Vc was used as the standard antioxidant. L9F was separately dissolved in distilled water to prepare flavonoid solutions of different concentrations (0.01, 0.02, 0.03... 0.09, and 0.10 mg/mL). Equal volume of DPPH solution was mixed with different concentrations of flavonoid solution. The mixture was shaken and then put in a dark place for 30 min. Finally, the absorbance was measured at 517 nm by a spectrophotometer and each experimental group had three parallel controls. Y%=1−A1−A2∕A3×100%

Y(%) is the DPPH radical scavenging activity A1 is the absorbance of the sample with DPPH, A2 is the absorbance of the sample without DPPH, and A3 is the absorbance of DPPH without the sample.

ABTS radical scavenging assay

The ABTS radical scavenging activities was collected as previously described in Kaur and Zhang (Kaur et al., 2020; Zhang, Yang & Zhou, 2018; Zhang et al., 2018). Vc was used as the standard antioxidant. L9F was separately dissolved in distilled water to prepare flavonoid solutions of different concentrations (0.01, 0.02, 0.03 ... 0.09, and 0.10 mg/mL). Two milliliters of ABTS solution was mixed with one hundred Microliters of different flavonoid solution. The mixture incubated in a dark place at room temperature for 6 min. Finally, the absorbance was measured at 734 nm by a spectrophotometer and each experimental group had three parallel controls. Y%=1−A1−A2∕A3×100%,

Y(%) is the ABTS radical scavenging activity, A1 is the absorbance of the sample with ABTS, A2 is the absorbance of the sample without ABTS, and A3 is the absorbance of ABTS without the sample.

Hydroxyl radical (•OH) scavenging assay

The hydroxyl radical scavenging activities of sample was collected as previously described in Chobot (Chobot & Hadacek, 2011), and Vc was used as the standard antioxidant. L9F was separately dissolved in distilled water to prepare flavonoid solutions of different concentrations (0.1, 0.2, 0.3 ... 0.9, and 1.0 mg/mL). A 0.5 mL flavonoid solutions of each concentration was mixed with 1 mL of Salicylic acid (6 mM), 1.5 mL of phosphate buffer solution (PBS, 0.15 M, pH 7.4), 1 mL of ferrous sulfate (6 mM) and 0.5 mL of H2O2 solution (0.01%). Then, the mixtures were incubated at 37 °C for 30 min. Finally, the absorbance was measured at 510 nm by a spectrophotometer and each experimental group had three parallel controls. Y%=A1−A2∕A3−A2×100%

Y(%) is the •OH radical scavenging activity (%), A1 is the absorbance of the sample after reaction with hydroxyl radicals, A2 is the absorbance of the sample, and A3 is the absorbance without H2O2.

Superoxide radical (•O2−) scavenging assay

The superoxide radical scavenging activities of sample was collected as previously described in Zhishen and Leong (Zhishen, Mengcheng & Jianming, 1999; Leong et al., 2008). Similarly, Vc was used as the standard antioxidant. L9F were separately dissolved in distilled water to prepare polysaccharide solutions of different concentrations 0.01, 0.02, 0.03 ... 0.09, and 0.10 mg/mL. A 1.0 mL sample of each concentration was mixed with 3.0 mL of Tris-Hcl buffer (pH 8.2), 0.8 mL of Pyrogallol (0.05 M). Then, the mixtures were bathed at 25 °C for 5 min and 1.0 mL HCl (8.00 M) was mixed after that. Finally, the absorbance was measured at 325 nm using a spectrophotometer. Y%=1−A1∕A2×100%

Y is the •O2− radical scavenging activity (%), A1 is the absorbance of the sample and A2 isthe absorbance of the control.

Results

The result of Single factor test

The influence of adsorption time

Figure 1 shows that the adsorption ratio increases first and then stabilizes with the increase of adsorption time. The adsorption ratio reached to the maximum first at 1 h,which was 59.06 ± 0.03%.

The influence of pH

Figure 2 shows that the adsorption ratio increases first, then goes down with the increase of pH. The adsorption ratio reach to the maximum at pH 3.5, which was (31.65 ± 0.03)%. There was no adsorption when pH was more than 5, which may be caused by the change of the flavonoid structure and the deactivation (Wu et al., 2017) or the weakening of the adsorption capacity of the macroporous resin (Li et al., 2007) under this pH condition.

The effect of temperature

Figure 3 shows that the adsorption ratio increases first and then goes down with the increase of temperature. The adsorption ratio reached to the maximum first at 30 °C, which was 51.26 ± 0.02%.

Optimization of extraction process by response surface methodology

Selection of analysis factor level

According to Box-Benhnken’s central combination test design principle and single-factor test results, three factors (temperature, pH, and time) that have significant effects on the extraction of total flavonoids were selected, and the three-factor three-level response is adopted, which is shown in Table 1.

Response surface analysis experiment design scheme

A (temperature), B (pH), and C (adsorption time) was taken as independent variables, and Y (total flavonoid adsorption ratio) was taken as the response value. The test plan and results are shown in Table 2.

Establishment and analysis of multivariate quadratic response surface regression model

Quadratic regression response surface analysis Performs in Table 2, and a multiple quadratic response surface regression model was established: Y = 0.47-0.056A-0.042B+0.031C+0.088AB-0.043AC+ 4 × 10−4BC-0.064A2-0.047B2+0.056C2. The variance analysis of each factor is shown in Table 3.

Table 3 shows that the model is significant (p < 0.05) and the Prob>F value of the decisive factor coefficient such as A (temperature), B (pH), C (adsorption time), AB (interaction between temperature and pH), AC (interaction between temperature and adsorption time) are 0.0008, 0.0039, 0.0161, 0.0004, 0.0192 (p < 0.05), indicating that the model has a good fit. In addition, the factors affecting the extraction ratio of total flavonoid were A (temperature), B (pH), and C (adsorption time) in order of magnitude, and the temperature reached a significant level (p < 0.001).

In this experiment, the interaction between AB and AC has significant effect. The results are shown in Figs. 4 and 5, and the interaction between BC is shown in Fig. 6. It can be seen from the response surface diagram that the extraction ratio of total flavonoids first increases and then decreases and increases in the end with the increase of A (temperature). B is in the range of −1.0 to 0. Due to the interaction of A, the extraction ratio of total flavonoids is higher. C is in the range of 0 to 1.0, due to the interaction of A, the extraction ratio of total flavonoids is higher.

Figure 1 Relationship between adsorption time and adsorption rate.

The data point indicates the change of adsorption rate over time.

Figure 2 Adsorption of D101 under different pH conditions.

The data point indicates the change of adsorption rate over pH.

Figure 3 Adsorption of D101 under different liquid temperature.

The data point indicates the change of adsorption rate over temperature.

Model verification

It is obtained that the supreme extraction ratio predicted is 64.2% under the process conditions of 25.02 °C, pH 2.80 and 1.85 h by solving the inverse matrix of the quadratic polynomial mathematical model of the total flavonoid yield.

The optimal conditions were revised to 25 °C, pH 2.80, and 1.85 h to extract the total flavonoids by the above to check the validity. The actual adsorption ratio was (64.14 ± 0.04)%, which is close to the theoretical value.

Table 1 Factors and the levels of experiment of Response Surface Analysis.

Factors	Factor levels	
	−1	0	1	
Temperature/°C	25	30	35	
pH	2.5	3.5	4.5	
time/h	0	1	2	

Table 2 Observed and estimated values for different levels of experimental design.

No.	Factors	Adsorption rate/%	
	A	B	C		
1	0	0	0	0.4739	
2	0	1	1	0.485	
3	0	0	0	0.4709	
4	1	0	−1	0.448	
5	0	−1	−1	0.4667	
6	1	1	0	0.3244	
7	1	−1	0	0.2403	
8	−1	−1	0	0.5625	
9	−1	0	−1	0.4396	
10	−1	1	0	0.2957	
11	0	0	0	0.4739	
12	0	0	0	0.4402	
13	0	1	−1	0.3885	
14	0	−1	1	0.5616	
15	−1	0	1	0.555	
16	0	0	0	0.4739	
17	1	0	1	0.3927	

Table 3 Analyze of mean square.

SS means sum of spuares; DF means degree of freedom; MS means mean square; F means a statistic obtained by analysis of variance based on experimental data; Prob > F means the chance that an F this large could occur due to noise.

Source	SS	DF	MS	F	Prob >F	
Model	0.1236	9	0.0137	17.2796	0.0005	
A-temperature	0.0250	1	0.0250	31.4807	0.0008	
B-pH	0.0142	1	0.0142	17.9143	0.0039	
C-time	0.0079	1	0.0079	9.9478	0.0161	
AB	0.0308	1	0.0308	38.7302	0.0004	
AC	0.0073	1	0.0073	9.1654	0.0192	
BC	0.0000	1	0.0000	0.0008	0.9782	
A ˆ2	0.0171	1	0.0171	21.5163	0.0024	
B ˆ2	0.0093	1	0.0093	11.7547	0.0110	
C ˆ2	0.0132	1	0.0132	16.6103	0.0047	
Error	0.0056	7	0.0008			
Lack of Fit	0.0047	3	0.0016	7.1415	0.0439	
Pure Error	0.0009	4	0.0002			
Total	0.1292	16				

The effect of eluent concentration

Figure 7 shows the ethanol elution effect with different volume fractions. As the volume fraction of the ethanol increases, the elution ratio is increasing. When 60% ethanol was used, the elution ratio of total flavonoids reached to (81.54 ± 0.03)%, and the purity of flavonoids was 7.13%. When 80% ethanol was used, although the elution ratio rises to (90.49 ± 0.03)%, the purity dropped to 3.61%. Therefore 60% ethanol is selected for elution considering the purity and elution ratio.

Figure 4 Response surface of interrelated influence of temperature and pH to flavonoids rate.

Figure 5 Response surface of interrelated influence of temperature and time to flavonoids rate.

Figure 6 Response surface of interrelated influence of pH and time to flavonoids rate.

Figure 7 Comparison of elution rate and purity of total flavonoids at different concentrations.

The data point indicates the Indicates the change of elution rate and purity of flavonoid over eluent.

The total flavonoids prepared by ethyl acetate extraction method, whose purity was 14.1%, but extraction ratio was only (31.68 ± 0.04)%, not only consumes a large amount of raw material and ethyl acetate, but also causes environmental problems. The macroporous resin adsorption method is more economical and environmentally friendly, and the eluate’s purity can rise to 13.69% with precipitated by absolute ethanol.

In vitro antioxidant activity

DPPH radical scavenging assay

DPPH has been widely accepted as a tool for estimating the free radical scavenging activities of antioxidants (Khled Khoudja, Boulekbache-Makhlouf & Madani, 2014). Figure 8 shows that the DPPH clearance ratio goes up as the concentration of each sample increases and it tends to be flat when the sample mass concentration is greater than 0.05 mg/mL. When the concentration reaches the maximum (0.1 mg/mL), the clearance ratio of total flavonoids on DPPH increases from (95.33 ± 0.01)% to (96.44 ± 0.04)% after purification, and Vc’s clearance ratio is (94.18 ± 0.002)%. It can be seen that the clearance effect of flavonoids on DPPH slightly better than that before purification, and the clearance effect of flavonoids on DPPH is close to Vc.

Figure 8 Scavenging ability of total flavonoids of endophytic fungi on DPPH.

Each data point indicates the clearance rate of total flavonoids on DPPH before and after extraction, where Vc is the control.

ABTS radical scavenging assay

ABTS has been widely accepted as a tool for estimating the free radical scavenging activities of antioxidants (Thaipong et al., 2012). Figure 9 shows that the ABTS clearance ratio goes up as the mass concentration of each sample increases. When the concentration reaches the maximum (0.1 mg/mL), the clearance ratio of total flavonoids on ABTS increases from (74.06 ± 0.04)% to (75.33 ± 0.03)% after purification, and Vc’s clearance ratio is (71.74 ± 0.05)%. It can be seen that the clearance effect of flavonoid on ABTS slightly better than that before purification, and the clearance effect of flavonoid on ABTS is better than Vc.

Figure 9 Scavenging ability of total flavonoids of endophytic fungi on ABTS.

Each data point indicates the clearance rate of total flavonoids on ABTS before and after extraction, where Vc is the control.

Hydroxyl radical (•OH) scavenging assay

Hydroxyl radicals are very active and have been associated with cancer risk when accumulated in the body excessively (Sakanaka, Tachibana & Okada, 2005). Figure 10 shows that the •OH clearance ratio goes up as the mass concentration of each sample increases and it tends to be flat when the sample mass concentration is greater than 0.3 mg/mL, but the clearance ratio of the sample before purification shows a downward trend, which may be the presence of oxidized and discolored impurities in the original fermentation broth. When the sample concentration reaches the maximum (1 mg/mL), the clearance ratio of total flavonoids on •OH increases from (3.98 ± 0.02)% to (73.79 ± 0.02)% after purification, and Vc’s clearance ratio is (93.72 ± 0.01)%. It can be seen that the clearance effect of flavonoids on •OH better than that before purification.

Figure 10 Scavenging ability of total flavonoids of endophytic fungi on −OH.

Each data point indicates the clearance rate of total flavonoids on −OH before and after extraction, where Vc is the control.

Superoxide radical (•O2−) scavenging assay

The superoxide radical is known to be very harmful to cellular components as a precursor of more reactive oxygen species, contributing to tissue damage and various diseases (Ozsoy et al., 2008; Aruoma, Grootveld & Bahorun, 2006). Figure 11 shows that the •O2− clearance ratio goes up as the mass concentration of each sample increases and it tends to be flat when the sample mass concentration is greater than 0.05 mg/mL, but the clearance ratio of the sample before purification shows a downward trend, which may be the presence of oxidized and discolored impurities in the original fermentation broth. When the sample concentration reaches the maximum (0.1 mg/mL), the clearance ratio of total flavonoids on •O2− increases from (0.15 ± 0.016)% to (28.11 ± 0.01)% after purification, and Vc’s clearance ratio is (31.14 ± 0.01)%. It can be seen that the clearance effect of flavonoids on •O2− better than that before purification.

Figure 11 Scavenging ability of total flavonoids of endophytic fungi on •O2−.

Each data point indicates the clearance rate of total flavonoids on •O2− before and after extraction, where Vc is the control.

Discussion

Flavonoids have attracted more and more attention because of their complex structure and function over these years (Baran et al., 2020; Zou et al., 2020; Zhang et al., 2020). However, flavonoids from plants are mainly used in the market now. The regeneration ratio is affected by the natural growth of plants, and the yield is limited. Therefore, the objective of this study was to extract flavonoid by macroporous resin and analyse antioxidant activities.

It has been shown that the extraction procedure has a significant impact on the yield and structural characteristics of flavonoids, as well as their biological activities (Taghinia, Khodaparast & Ahmadi, 2019). The macroporous resin adsorption method has been widely used in the extraction and purification of total flavonoid from plants (Du et al., 2012; Zhang et al., 2018). The macroporous resin adsorption method is more economical and environmentally friendly with similar purity and higher extraction ratio compared with commonly used organic solvent method.

It has been found that the generation of reactive oxygen species (ROS) and the corresponding response to oxidative stress are critical factors in the outbreak of several human diseases (Lee & Lee, 2006). Antioxidants have vital functions against ROS in the biological system (Zhang et al., 2015). In the present study, the antioxidant activity of flavonoid was studied by DPPH, ABTS, superoxide radical and hydroxyl radicals. The results showed that flavonoids exhibited stronger antioxidant activity than against DPPH, ABTS, superoxide radical and hydroxyl radicals, and the clearance ratio is highly closed to total flavonoid isolated by ethyl acetate (Tang et al., 2020).

These results indicated that the total flavonoid extracted from the endophytic fungus L9 from Conyza blinii H. Lév by macroporous resin has good antioxidant activity, which is closed to ethyl acetate extraction. The method with macroporous resin is better than which with ethyl acetate in many aspects. Further scientific work in our laboratory is in progress to separate it.

Conclusions

In the present study, we used the response surface method to optimize the extraction of total flavonoids from the endophytic fungus L9 from Conyza blinii H. Lév by macroporous resin for the first time. It was found that the best adsorption ratio reached (64.14 ± 0.04)% and the purity of the total flavonoids was increased by 14.55 times compared with the original fermentation broth. In the antioxidant research, L9F-M has good antioxidant activity in vitro. These results demonstrate that macroporous resin in the extraction of total flavonoids from endophytic fungus is better than organic solvents with higher extraction ratio, safety, lower cost and good antioxidant, which provides more evidence for the production of flavonoids by biological fermentation method. However, the further applications remain to be explored in future studies.

Supplemental Information

File S1 The dataset of the single factors and anti-oxidation

The single factor is showing the absorbance and absorbance rate of the sample in related experiments. These data are used to compare the changes in the purity and activity of total flavonoids before and after extraction. The anti-oxidation is showing the clearance of the sample to DPPH ABTS Hydroxyl and Superoxide.

Click here for additional data file.

Additional Information and Declarations

Competing Interests

Author Contributions

Data Availability

The authors declare there are no competing interests.

Shuheng Zhao conceived and designed the experiments, performed the experiments, prepared figures and/or tables, and approved the final draft.

Xulong Wu performed the experiments, analyzed the data, prepared figures and/or tables, and approved the final draft.

Xiaoyu Duan, Caixia Zhou and Zhiqiao Zhao performed the experiments, prepared figures and/or tables, and approved the final draft.

Hui Chen analyzed the data, authored or reviewed drafts of the paper, and approved the final draft.

Zizhong Tang conceived and designed the experiments, authored or reviewed drafts of the paper, and approved the final draft.

Yujun Wan, Yirong Xiao and Hong Chen analyzed the data, authored or reviewed drafts of the paper, and approved the final draft.

The following information was supplied regarding data availability:

The raw measurements are available in the Supplemental File.

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
