# Peer review of "Optimal extraction, purification and antioxidant activity of total flavonoids from endophytic fungi of Conyza blinii H. Lév"

_PeerJ, doi:10.7717/peerj.11223_

## Round 0.1 · original submission · Major Revisions

Please pay particular attention to the comments on the methodology.

Reviewer 1 ·

Basic reporting

The PeerJ Manuscript 55486v1. A conceptual mistake, the fungus is not a plant as described in section 2.1 (lines 62 and 63). The species name of the fungus must be provided (line 19); it’s ambiguous in some of the Methods described that make impossible to repeat: …was higher than before… (line 32), the meaning it´s not clear; all the Keyword are included in the title (line 39); …grew to a certain size (lines 75 and 76), a wet or dry weight must be provided; a small amount (line 80) specify a volume value; …and repeated three times like that (line 82), if this is an extraction, it is not clear how did they do the separation; …with slight modifications (line 86), without a explanation of the modifications; …with certain… an amount must be provided (line 91); it´s not clear the meaning of 2.3.5 or 2.4 on line 105 and 106; need the Absorption rate units in figures 1 to 6; need the Elution rate and Purity units in figure 7; in figures 9 to 11 the Clearance rate units must be given.
The English must be improved: …the regeneration speed.. (line 18), must be the production speed…; …is an Flavonoid… (lines 19 and 52), must be …is a flavonoid…; 60% ethanol was the best elution volume (line 26) must be …elution solvent; there are several type writing spaces, like in line 31, 39, 44, 87, 90, 177 and 186; …for further effect… (line 31), must be …for further uses…; …providing more theoretical basis for… (line 37) must be … evidence for…; …endophytic fungi… (line 52), must be endophytic fungus..; …total flavonoid… (line 58), must be …total flavonoids…; …then extract… (line 91), must be …then the mixture was incubated…; …and elution rate… (line 104) must be and elution time..; the Tables 1, 2 and 3 have two points instead of only one (lines179, 183 and 189); …optimiza… (line 299) ,,,optimize…; Figure 3 …under different temperature to liquid, must be …under different liquid temperature…

Experimental design

The paper is original and is within the aims and scope of the journal. The research question is well defined, but methods are not well described and ambiguous that make it hard to reproduce.

Validity of the findings

There is not data to support the purification times an important value to evaluate the good performance of the method.

The conclusion section is a result resume, don't have any conclusion

Additional comments

The English must be revised and more specific data have to be provided in order to be able to reproduce the method,

Reviewer 2 ·

Basic reporting

The English should be carefully revised. Some sentences are hard-to-read. The other points are OK.

Experimental design

It is not clear the gap that is going to be filled with this research article. It is difficult to understand how the authors establish that the extracted content is mainly flavonoids.

Validity of the findings

Although it is an interesting research article, I think there are a few details that should be clarified, and some conclusions are not evident from the described results.

Additional comments

The manuscript is interesting, and new methodologies to extract flavonoids are welcomed. However, some points should be clarified before the article's acceptance.
It is difficult to understand the evaluated extracts; for example, the L9F-E was not assed?
How was obtained the extract L9F-M?
The sentence from line 104 to 106 is confusing.
Why use ethyl acetate for the extraction? Some flavonoids are less soluble in this solvent.
How was the extract's flavonoid content established?
The authors should clarify how the research provides a more theoretical basis for producing flavonoids by biological fermentation method?

---

## Round 0.2 · Minor Revisions

The manuscript was significantly improved but still contains minor points to address. Please see the Reviewer's comments to address these minor points.

Reviewer 1 ·

Basic reporting

All my comments or answers were addressed in the file peerj-55486-Response_letter_to_peerj.docx, but some are deficient in the paper peerj-reviewing-55486-v1.pdf.
For example:
The keywords were substituted for similar others, that still in the title.
There still some mistakes as in Figure 7: The data point indicate the Indicate ….
Point 10 of my letter, if Elution rate is a %, they do not express that in the figure.
Response 10, It represent the proportion of sample in the population. This can not be an Elution rate.

Experimental design

Still not clear:
line 70: .. until the mycelial pellets grew to a certain condition (Bacteria is not growing and The fermentation broth is dark enough). I still thinking that a mycelium dry weight must be provided.
line 75: A small amount of concentrated fermentation broth (mL) was mixed… The amount must be given.

Validity of the findings

Table 3, they do not give the meaning of SS, DF, MS, F and >F, so for the Table is not clear.

Additional comments

I still thinking that the manuscript could be improved English and some data must be provided.

---

## Round 0.3 · accepted · Accept

The manuscript is now suitable for publication.